# OpenReview forum: "New Job, New Gender? Measuring the Social Bias in Image Generation Models"
_acmmm.org/ACMMM/2024/Conference — MM2024 Oral_

### Official Review · Reviewer_EMDo · 2024-05-07

**Rating:** 4
**Confidence:** 3

**Summary:**

This paper presents BiasPainter, a new evaluation framework for measuring social biases in image generation models. Experiments on six image generation models show that BiasPainter effectively identifies and quantifies biased behavior with high accuracy. Additionally, its potential to reduce bias in these models is demonstrated.

**Strengths:**

1. This work is working on social bias detection and mitigation in image generation, an essential task for generative AI society. The sensitive attributes, such as gender, age, and race, are considered.
2. An evaluation framework is proposed. This could help us to underline those biases in other application models with the same metrics for better comparison.
3. Six image generation models are applied, and a strong and concrete evaluation is done.
4. The clarity of the paper is good from my perspective.

**Limitations:**

1. Line 924 has an additional ‘.’ after the word BiasPainter.
2. In gender and age assessment, from my point of view, it is better to compare more than one API/model age estimation results instead of just using one single API to claim the subject’s age. Since the estimation results could not be validated, do the authors try human manual evaluation or another way to ensure this assessment is reliable?
3. A relevant and reasonable image processing pipeline determines the skin tone color score. Still, there is a common issue in image processing: the influence of different lighting conditions in every image. I saw that the authors do not consider the highlighted region of the face, but the issue I am concerned about is the color intensity or brightness of the image itself. Do you consider this matter? This is a promising pipeline, anyway.
4. ‘Generally, the Stable Diffusion Models are more biased than the other three models. Stable-Diffusion 2.1 is the most biased model on age, and Pix2pix shows less bias on age and gender.’ for this sentence, can you provide insights or any comments on why this happened?
5. For Table 4, from this paper's clarification, does it mean that the more the value of the word bias score is towards 0, the more likely it is not to be so biased? From my understanding, is the bias mitigation done by balancing the dataset class distribution of that particular target variable with the knowledge of the RQ2 outcome? How do you prove that it is your RQ2 that contributes to the decrease in score? we all knew that balancing the classes would decrease this score since we have more balanced training data.

**Suitability:**

3

---

### Official Review · Reviewer_zRLh · 2024-05-24

**Rating:** 4
**Confidence:** 2

**Summary:**

This paper introduces "BiasPainter," an evaluation framework designed to automatically detect social biases in image generation models. The framework employs a broad range of queries targeting various sensitive biases and assesses numerous contemporary generative models.

**Strengths:**

1. The evaluation coverage is broad, including 62 professions, 39 activities, 57 types of objects, and 70 personality traits.
2. The evaluated models are cutting-edge generative models, including StableDiffusion, Midjourney, et al.
3. The results, as presented in Table 1, are detailed and comprehensive, offering a clear overview of model biases.

**Limitations:**

1. While the evaluation results are detailed, some of the metrics used in Section 3.5 seem overly reliant on manual techniques.  For example, the race bias score is calculated using grayscale values to estimate skin color, which may not yield accurate results and could affect the robustness of the evaluation framework.

2. The framework requires both a seed image and a text prompt for input. This requirement limits its applicability since many popular online image generation systems do not support image inputs. How to expand the evaluation target to a broader range of models remains a problem.

**Suitability:**

2

---

### Official Review · Reviewer_uAXb · 2024-06-05

**Rating:** 5
**Confidence:** 1

**Summary:**

The paper presents BiasPainter, a framework to detect social bias in generated images. It analyzes 3 main categories of bias: gender, race, and age.
BiasPainter introduces a novelty approach to detect bias in generated content: it compares the generated images with ground truth references to acquire the bias score, while previous methods just leverage the prompted text.

**Strengths:**

The automatic bias detection process helps in mitigating the requirements for human effort. This makes the solution scalable and potentially broadens the application of social bias detectors.

**Limitations:**

The categories considered in this work only cover a narrow distribution of social bias in generated images. The seed image set used as a baseline to detect social bias consists of photos manually selected from the VGGFace2 dataset, where 3 photos from each combination are selected, thus obtaining a total of 54 photos.
Using a wider population for the seed set, the distribution of considered classes might be better addressed.
Moreover, the collection process for the seed set is dependent on the bias of the human operator.

**Suitability:**

2

---

### Meta-Review · Area_Chair_616Z · 2024-07-03

**Recommendation:** Accept (Oral)
**Confidence:** 4

**Metareview:**

The paper introduces a timely framework for detecting social biases in image generation models. The reviews highlight several strengths, including the framework’s scalability, comprehensive coverage, and robust evaluation. However, there are also noted limitations, such as the narrow scope of biases considered and potential methodological issues.

After considering that:
i) the framework proposed in this paper offers a significant advancement in the detection and mitigation of social biases in generated images;
ii) the paper’s strengths outweigh its limitations;
iii) the authors’ effective response to feedback, providing additional experiments and clarifications (that should be added to the paper).
I believe that this paper can make a valuable addition to the conference. Therefore, I recommend acceptance of this paper.